# PUMA: Empowering Unified MLLM with Multi-grAnular Visual Generation

a) The Diversity and Controllability Requirements of Different Visual Generative Tasks

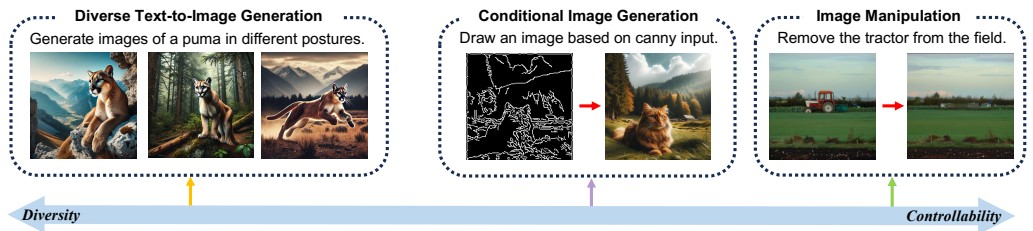

b) PUMA for Various Visual Generation and Understanding Tasks

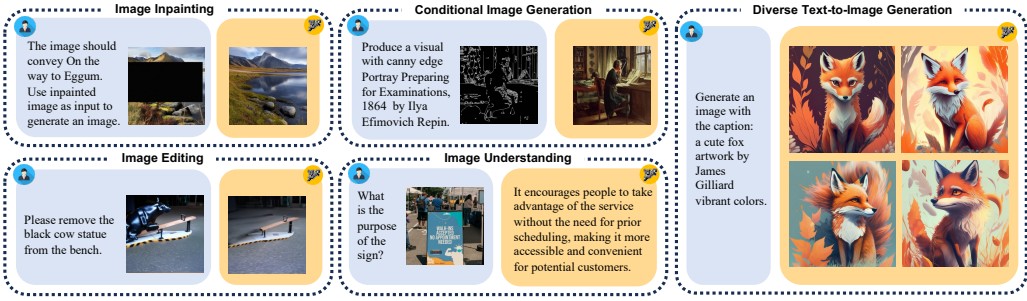

Figure 1: **a)** Diversity and controllability tradeoff in image generation tasks: diverse text-to-image generation requires high diversity and fidelity, while tasks like conditional generation and manipulation require high controllability on the image. **b)** The introduced PUMA, a unified multimodal large language model that processes and generates multi-granular visual representations, balancing diversity and controllability across visual generation tasks. It excels in image understanding, diverse text-to-image generation, editing, inpainting, colorization, and conditional image generation.

## Abstract

Recent advancements in multimodal foundation models have yielded significant progress in vision-language understanding. Initial attempts have also explored the potential of multimodal large language models (MLLMs) for visual content generation. However, existing works have insufficiently addressed the varying granularity demands of different image generation tasks within a unified MLLM paradigm — from the diversity required in text-to-image generation to the precise controllability needed in image manipulation. In this work, we propose **PUMA**, em**P**owering **U**nified MLLM with **M**ulti-gr**A**nular visual generation. PUMA unifies multi-granular visual features as both inputs and outputs of MLLMs, elegantly addressing the different granularity requirements of various image generation tasks within a unified MLLM framework. Following multimodal pretraining and task-specific instruction tuning, PUMA demonstrates proficiency in a wide range of multimodal tasks, including image understanding, diverse text-to-image generation, editing, inpainting, colorization, and conditional generation. This work represents a significant step towards a truly unified MLLM capable of adapting to the granularity demands of various visual tasks. The code and model will be released upon acceptance.

# 1 INTRODUCTION

Unifying multimodal understanding and generation capabilities within a single model is a critical milestone toward artificial general intelligence (AGI). Towards this goal, recent advancements (Liu et al., 2024b; Zhu et al., 2023a) in multimodal large language models (MLLMs) have made significant progress in integrating visual reasoning and understanding with natural language interfaces. However, developing a unified framework that excels at both comprehending and generating multimodal content remains a significant challenge in the field of artificial intelligence.

Recent studies (Sun et al., 2023; Ge et al., 2024b) have explored MLLM's potential for visual generation, beyond the previously well-explored visual understanding and reasoning with MLLMs. These approaches enable MLLMs to process image-text inputs and produce either textual outputs or semantic-level visual tokens. In the case of image generation, these visual tokens are subsequently transformed into pixel-space images using diffusion-based decoders. Such unified frameworks empower MLLMs to perform a wide spectrum of tasks within a single framework, ranging from detailed visual analysis to creative image synthesis.

However, existing MLLM-based methods (Sun et al., 2023; 2024b) face a common challenge in the trade-off between diversity for text-to-image generation and high controllability for tasks such as image editing. Previous methods mostly rely on single-granular features extracted from a visual encoder and neglect the varying granularity requirements of different tasks. On the one hand, generating diverse images reflecting the real world from text descriptions requires features that encode coarse semantic concepts. Such features are fed as conditions into the diffusion-based image decoder, allowing the diffusion model to generate diverse images that semantically align with the text prompt. On the other hand, tasks demanding precise control over output images, such as image editing and inpainting, require the LLMs to predict fine-grained features that encode rich, detailed visual information for the image decoder. This dichotomy presents a significant challenge for current MLLM-based methods, which typically generate single-granular feature representations for all tasks. As a result, models optimized for diverse image generation often lack the fine-grained controllability necessary for detailed downstream tasks such as editing, while those focused on precise controllability produce less varied outputs for the task of text-to-image generation. Although recent work like SEED-X (Ge et al., 2024b) attempts to bypass this issue by leveraging condition images directly input to the diffusion-based decoder for fine-grained control, a unified solution to the multi-granularity problem remains underexplored.

Towards the multi-granular feature demands of various tasks, we propose a novel paradigm em**P**owering **U**nified MLLM with **M**ulti-gr**A**nular visual generation (**PUMA**). PUMA facilitates seamless integration of image generation and understanding processes, while simultaneously handling multiple feature granularities — from coarse-grained abstractions to fine-grained details — within a single framework. By leveraging multi-scale features, our approach empowers MLLMs to excel in diverse image generation and controllable downstream tasks, within a unified framework.

Our method comprises three key modules: 1) An image encoder that extracts multi-granular representations, which serve as the foundation for visual generation and understanding; 2) An autoregressive MLLM that processes and progressively generates multi-scale image features; and 3) A set of dedicated diffusion-based image decoders that decode images from MLLM-generated features at multiple granularities. To optimize this framework, we employ a two-stage training strategy: first fine-tuning the set of pre-trained diffusion models as our image decoders, where each model reconstructs or generates images conditioned on the corresponding feature granularities from the encoder; then training the autoregressive MLLM with regression loss supervised by the multi-scale encoder features to process and generate multi-granular image features. PUMA leverages large-scale pre-training followed by task-specific instruction tuning on a collection of linguistic-visual datasets, enabling our model to handle various tasks including image understanding, text-to-image generation, editing, inpainting, colorization, and conditional generation.

In summary, we introduce a novel multi-granularity paradigm for MLLMs that addresses the limitations of existing single-scale methods. By simultaneously processing and generating features at multiple granularities, our approach enables a unified framework to handle a wide range of tasks, from diverse image generation to precise editing and highly controllable generation. This unified framework represents a significant advancement towards more versatile and capable MLLMs, contributing to the broader goal of achieving AGI in multimodal domains.

## 2 RELATED WORK

### 2.1 MULTIMODAL UNDERSTANDING

The rapid advancement of large language models (LLMs) has catalyzed significant progress in multimodal large language models (MLLMs) for multimodal understanding tasks Dai et al. (2023); Li et al. (2024b); Zhang et al. (2023a); Chen et al. (2024); Lin et al. (2024); Zhang et al. (2024b); Li et al. (2024a). Pioneering works such as LLaVA (Liu et al., 2024b) and MiniGPT-4 (Zhu et al., 2023a) have demonstrated remarkable performance across diverse image understanding tasks, including visual question answering (VQA), visual reasoning, optical character recognition (OCR), and object grounding. These approaches typically employ visual encoders, such as the CLIP encoder (Radford et al., 2021), to extract continuous image features, which are then projected into the LLM's embedding space for subsequent tasks. While successfully unifying various image understanding tasks within a single model, these methods mostly adhere to a multimodal-input, text-output paradigm. Consequently, they excel at text-based responses to visual inputs but cannot generate multimodal outputs beyond text, limiting their applicability in tasks requiring visual content generation.

### 2.2 UNIFIED UNDERSTANDING AND GENERATION FOR MLLMS

Recent research has focused on equipping MLLMs with multimodal output capabilities (Wu et al., 2023; Tang et al., 2024; Ye et al., 2024a; Zhu et al., 2023b). GILL (Koh et al., 2024) pioneered the integration of image generation abilities into MLLMs. Subsequently, SEED-LLaMA (Ge et al., 2023) and Emu (Sun et al., 2023) further advanced image generation and understanding capabilities within MLLMs, while DreamLLM (Dong et al., 2023) proposed an end-to-end training approach for enhanced performance.

More recent works, such as SEED-X (Ge et al., 2024b) and Emu2 (Sun et al., 2024b), have scaled up MLLMs for unified generation, adopting continuous feature-based methods. These approaches utilize pre-trained vision encoders to extract continuous semantic features, which MLLMs then autoregressively regress. Specialized diffusion model-based decoders transform these MLLM-generated features into pixel-space images. However, the single-scale image feature generation pipeline employed by these methods struggles to address tasks with varying granularity demands, making it challenging to balance diverse image generation with fine-grained control for manipulation tasks. SEED-X attempts to address the multi-granularity issue by introducing conditional image input to the diffusion-based decoder for fine-grained control. However, this approach limits its applicability to image editing tasks encountered during decoder training. Consequently, a unified solution to the multi-granularity problem remains underexplored. In contrast, our work proposes a novel multi-granularity paradigm that addresses these limitations by simultaneously handling multiple levels of feature granularity within a single, unified framework.

Alternative approaches have also been investigated. Chameleon (Team, 2024) explored using discrete image tokens to bridge image understanding and generation, but the vector quantization process leads to information loss, hindering high-performance image understanding. TransFusion (Zhou et al., 2024) and show-o (Ge et al., 2023) proposed transforming the MLLM backbone itself into a denoiser in a diffusion-based or demasking-based approach. However, these methods require numerous denoising steps for each image generation, resulting in substantial computational costs given the scale of current MLLM backbones. VAR (Tian et al., 2024) is another track of generation framework that implements hierarchical autoregressive with discrete tokens for image generation, but it only discusses image generation and cannot unify multimodal tasks.

## 3 METHOD

Existing approaches typically optimize for either fine or coarse-grained features, resulting in a trade-off between precise control and generation diversity. To overcome this limitation, we propose **PUMA**, a unified multi-granular MLLM paradigm. Our approach simultaneously processes multiple levels of feature granularity within a unified MLLM framework, facilitating seamless transitions across a wide spectrum of multimodal tasks.

Our framework consists of three key components: an image encoder (Sec. 3.1), a set of image decoders conditioned on different granular features (Sec. 3.2), and a multi-granular autoregressive MLLM (Sec. 3.3). These components work synergistically to extract, process, and generate multi-scale image features, adapting to various task-specific granularity requirements. To optimize our MLLM, we employ a two-stage process of pretraining and instruction tuning (Sec. 3.4), enabling it to perform a wide range of tasks including image understanding, generation, editing, and conditional image generation.

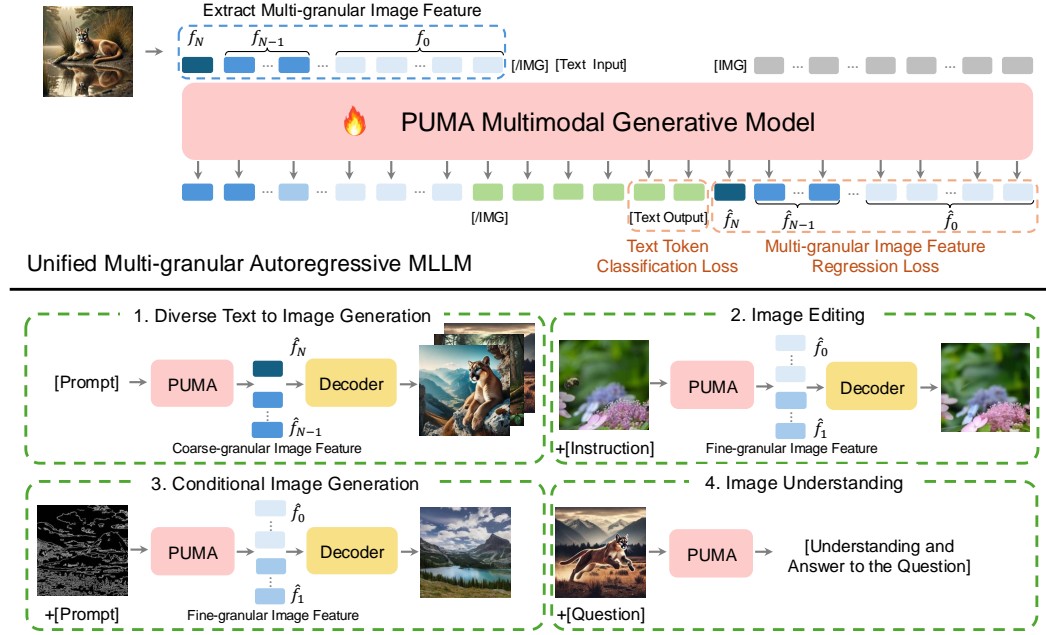

Figure 2: **Upper:** PUMA's unified multi-granular autoregressive pipeline for processing and generating text and multi-granular visual features. **Lower:** Illustration of PUMA's versatility across various tasks: 1) diverse text-to-image generation, 2) image editing, 3) conditional image generation, and 4) image understanding, showcasing different input-output configurations.

## 3.1 IMAGE ENCODING AND MULTI-GRANULAR FEATURE EXTRACTION

Our unified multi-granularity paradigm leverages a semantic image encoder to extract multi-scale features, forming the foundation for diverse visual task processing. We employ a CLIP (Radford et al., 2021) semantic image encoder to process input images $x$ and generate the initial set of high-resolution features $f_0 \in \mathbb{R}^{H \times W \times C}$, with $H$ and $W$ representing the spatial dimensions of the highest resolution feature grid, and $C$ denoting the channel dimension. In our setting, the feature size is $H = W = 16$, thus the highest resolution feature $f_0$ has 256 visual tokens.

To obtain multi-granular representations, we derive lower resolution features through successive applications of 2D average pooling with kernel size 2 and stride 2:

$$f_i = \text{AvgPool}(f_{i-1}), \quad i = 1, 2, ..., N \tag{1}$$

where $N$ is the number of additional granular levels. This process generates a series of feature grids at progressively coarser resolutions, ranging from fine-grained features preserving detailed spatial information and local textures, through mid-level features capturing object parts and regional structures, to features representing coarse-grained semantic concepts. These features are denoted as $f_0, f_1, f_2, f_3,$ and $f_4$, which have 256, 64, 16, 4, and 1 visual tokens respectively.

## 3.2 MULTI-GRANULAR VISUAL DECODING

Image features at different granularities encode varying levels of information. We employ diffusion-based models as decoders due to their flexible capability to handle multi-scale features. When processing coarse-grained semantic features, the decoders can effectively synthesize missing fine-grained information with their learned image priors and generate diverse, semantics-aligned images. On the other hand, when handling fine-grained features, they accurately reconstruct precise image details. This versatility in generating or reconstructing images across different granularities makes diffusion-based models suitable for our multi-granularity approach.

We develop a set of dedicate diffusion-based image decoders $D_0, D_1, ..., D_N$ corresponding to the feature scales $f_0, f_1, ..., f_N$. These decoders enable the visual decoding of images at various levels of granularity. We formulate the image decoding process for each granularity level $i$ as $\hat{x}_i = D_i(f_i, z)$, where $\hat{x}_i$ is the decoded image, $f_i$ is the feature map at granularity level $i$, and $z$ is a random noise vector for the diffusion process.

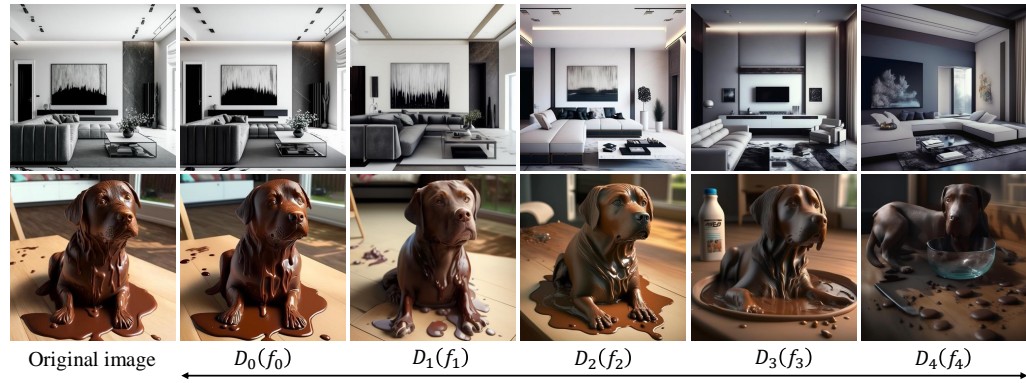

Original image    $D_0(f_0)$    $D_1(f_1)$    $D_2(f_2)$    $D_3(f_3)$    $D_4(f_4)$

Fine-grained reconstruction      Semantic-guided generation

Figure 3: Multi-granular visual decoding from fine-grained to coarse-grained granularity.

We leverage the pre-trained SDXL models (Podell et al., 2023) as our decoding framework and fine-tune these pre-trained models to generate or reconstruct images conditioned on different granular features. By modifying the conditional input mechanism through cross-attention in SDXL to accept our multi-granular features $f_i$, we harness the models' inherent ability to decode coherent images.

Fig. 4 shows the training process of different granular image decoding, during which the image encoder is frozen to preserve semantic property. Fig. 3 illustrates the visual decoding capabilities of multi-granular decoders. The visualizations demonstrate the fidelity of decoded images across different granularities, with finer-grained features yielding reconstructions closer to the original input, and coarser-grained features leading to image generation guided by the semantics of the input image. This validates the effectiveness of our approach in preserving and utilizing multi-granular visual information.

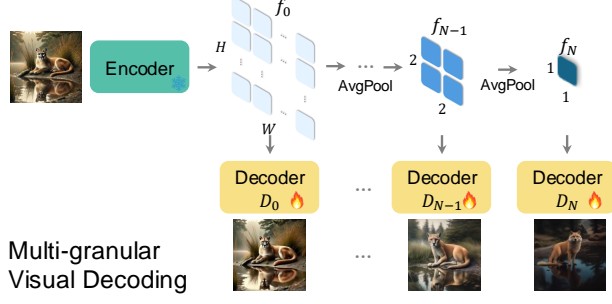

Figure 4: Training phase of multi-granular visual decoding.

This multi-granular decoding framework, in conjunction with our hierarchical feature extraction, establishes a foundation for the subsequent stages of our MLLM architecture, paving the way for diverse visual tasks in later training phases.

### 3.3 PROGRESSIVE MULTI-GRANULAR IMAGE MODELING IN AUTOREGRESSIVE MLLM

Driven by the goal of utilizing a unified framework capable of adapting to a wide range of visual-linguistic tasks with varying granularity requirements, we design an autoregressive MLLM to process and generate both text tokens and multi-granular image features.

Our autoregressive MLLM, denoted as $M$, processes text and multi-granular image features progressively, as illustrated in Fig. 2. The model processes features token by token, predicting each token sequentially within each granularity level, and progressing from the coarsest level $N$ to the finest level $0$. This approach allows the model to refine its predictions as more detailed information becomes available.

We structure the input sequence as a concatenation of text tokens and flattened image feature tokens from multiple granularity levels. This progressive approach enables the model to capture dependencies across different scales, from coarse global structures to fine local details.

The MLLM is trained using an autoregressive next token prediction objective, combining both text and image losses:

$$\mathcal{L} = -\sum_i \log P(t_i | t_{<i}, F_{<i}) + \sum_{i=0}^{N} \alpha_i \sum_{j=1}^{k_i} |f_{i,j} - \hat{f}_{i,j}|^2 \tag{2}$$

The first term represents the cross-entropy loss for text token prediction, where $t_i$ are text tokens. The second term is the regression loss for image feature prediction, where $f_{i,j}$ and $\hat{f}_{i,j}$ are the

ground truth and predicted feature tokens, respectively, at the $i$-th granularity level. $k_i$ is the number of visual tokens at the $i$-th granularity level. The coefficient $\alpha_i$ allows for adjusting the importance of each granularity level during training.

### 3.4 MULTIMODAL PRETRAINING AND INSTRUCT TUNING

To demonstrate the effectiveness of our unified multi-granularity paradigm, we implement a comprehensive two-stage training pipeline for PUMA: multimodal pretraining followed by task-specific instruct tuning. This approach allows our model to first acquire broad multimodal capabilities before specializing in targeted visual-linguistic tasks during the subsequent instruct tuning stage.

**Multimodal Pretraining:** Our multimodal pretraining leverages a diverse set of large-scale datasets: Laion-2B (Schuhmann et al., 2022), Laion-Aesthetics (Burger, 2023), GRIT (Peng et al., 2023), The Pile (Gao et al., 2020), OCR-VQA-200K (Mishra et al., 2019), and LLaVAR (Zhang et al., 2023b). This combination of datasets provides a rich mixture of image-text pairs, textual data, and specialized visual question-answering samples. To enhance the model's bidirectional understanding of image-text relationships, we employ a dynamic training strategy that randomly alternates between text-to-image and image-to-text tasks for each image-text pair.

**Instruct Tuning:** Following pretraining, we conduct targeted instruct tuning to adapt our model to specific visual-linguistic tasks. To evaluate PUMA's performance across different task types, we fine-tune four dedicated models for the four types of tasks, each initialized from the pretraining checkpoint.

*High-quality Text-to-Image Generation:* We utilize Laion-Aesthetics (Burger, 2023) and JourneyDB (Sun et al., 2024a) to focus on generating aesthetically pleasing and diverse images.

*Precise Image Manipulation:* Training on the SEED-Edit (Ge et al., 2024a) dataset enables accurate and controlled image editing.

*Conditional Image Generation:* The subset of MultiGen-20M dataset (Qin et al., 2023) including canny-to-image, inpainting, and colorization is employed to equip the model with the ability to generate images under specific conditions and constraints.

*Image Understanding:* Fine-tuning on the subset of LLaVA-OneVision (Li et al., 2024a) and Cambrain (Tong et al., 2024) to enhance the model's image comprehension capabilities. Data about math/reasoning and cross-duplicated data in the two datasets are removed.

## 4 EXPERIMENTS

We present our experimental results as follows: Sec. 4.1 details our experimental setup. In Sec. 4.2, we evaluate the effectiveness of our multi-granularity feature encoding and diffusion-based multi-granularity image decoders. We then demonstrate PUMA's versatility across various tasks: diverse text-to-image generation (Sec. 4.3), image editing (Sec. 4.4), conditional image generation (Sec. 4.5), and vision-language understanding (Sec. 4.6).

### 4.1 SETUP

Our unified multi-granular MLLM employs LLaMA-3 8B (Touvron et al., 2023) as the language model backbone and CLIP-Large ($224 \times 224$ input) (Radford et al., 2021) as the image encoder. The image decoders are initialized from pretrained SDXL models (Podell et al., 2023). For more details on the experimental setup, please refer to the Appendix.

### 4.2 MULTI-GRANULAR VISUAL DECODING

We evaluate the multi-granular visual decoding capabilities of our model using multi-scale features from the encoder (Sec. 3.1) and dedicated visual decoders (Sec. 3.2). Our aim is twofold: to achieve precise reconstruction using fine-grained feature scales (such as $f_0$ and $f_1$), and to implement high diversity semantics-guided image generation using coarse-grained features (such as $f_4$ and $f_3$). It is worth mentioning that in this subsection we validate the multi-granularity encoder and decoders (Fig. 4), while the MLLM (Sec. 3.3) is not leveraged for the experiments in this subsection.

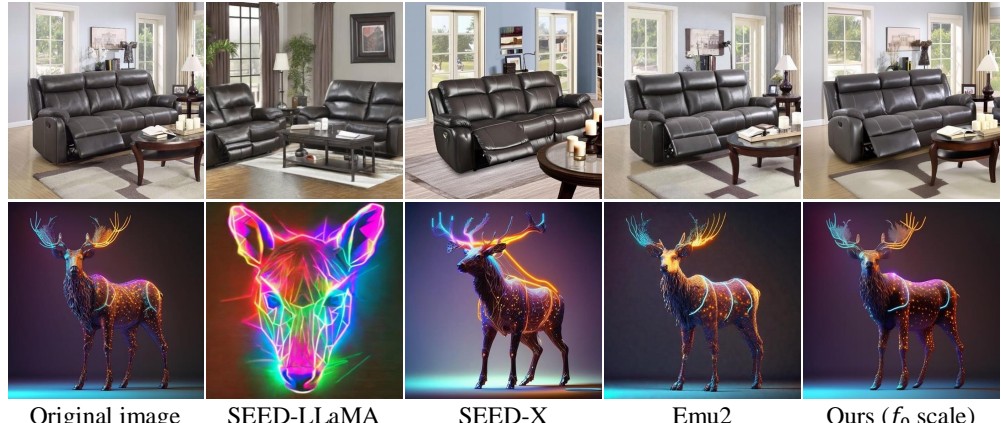

Original image     SEED-LLaMA     SEED-X     Emu2     Ours ($f_0$ scale)

Figure 5: Fine-grained image reconstruction of SEED-LLaMA (Ge et al., 2023), SEED-X (Ge et al., 2024b), Emu2 (Sun et al., 2024b) and PUMA ($f_0$ scale). High quality image reconstruction is the foundation of precise image manipulation tasks.

Table 1: Image decoding evaluation using image encoder and decoder on the ImageNet validation set. $\text{PSNR}_r$ and $\text{LPIPS}_r$ measure the difference between reconstructed and ground truth images. $\text{PSNR}_d$ and $\text{LPIPS}_d$ measure the difference between two separate reconstructions of the same image, reflecting decoding diversity.

| Model | Encoder foundation | Token num. | $\text{PSNR}_r\uparrow$ | $\text{LPIPS}_r\downarrow$ | $\text{PSNR}_d\downarrow$ | $\text{LPIPS}_d\uparrow$ |
|---|---|---|---|---|---|---|
| SEED-LLaMA (2023) | BLIP-2 ViT (0.3B) | 32 | 9.73 | 0.6756 | **10.45** | **0.6189** |
| SEED-X (2024b) | Qwen-VL Encoder (4B) | 64 | 10.86 | 0.5152 | 11.60 | 0.4292 |
| Emu2 (2024b) | EVA02-CLIP-E-plus (4B) | 64 | 15.72 | 0.2532 | 16.07 | 0.2101 |
| PUMA ($f_4$ scale) | CLIP-Large (0.3B) | 1 | 10.76 | 0.6481 | 12.82 | 0.5751 |
| PUMA ($f_3$ scale) | CLIP-Large (0.3B) | 4 | 11.04 | 0.5971 | 12.61 | 0.5329 |
| PUMA ($f_2$ scale) | CLIP-Large (0.3B) | 16 | 12.35 | 0.4992 | 13.50 | 0.4354 |
| PUMA ($f_1$ scale) | CLIP-Large (0.3B) | 64 | 13.26 | 0.4325 | 14.12 | 0.3631 |
| PUMA ($f_0$ scale) | CLIP-Large (0.3B) | 256 | **18.16** | **0.2215** | 19.36 | 0.1559 |

### 4.2.1 FINE-GRAINED IMAGE RECONSTRUCTION

Fine-grained image reconstruction is crucial for preserving image details, yet it has posed significant challenges for models like SEED-LLaMA (Ge et al., 2023), SEED-X (Ge et al., 2024b), and Emu2 (Sun et al., 2024b). While SEED-LLaMA and SEED-X struggle with detailed reconstruction, limiting their precise image manipulation capabilities without additional techniques such as conditional image input (as used in SEED-X), Emu2 attempts to improve reconstruction by scaling up its image encoder to 4 billion parameters. Our approach achieves superior reconstruction quality with a more efficient architecture. We employ the CLIP-Large encoder (0.3 billion parameters), which is over 10 times smaller than Emu2's, and implement fine-grained level image embedding with 256 tokens. As demonstrated in Tab. 1, our method using $f_0$ scale features achieves 18.16 $\text{PSNR}_r$ and 0.2215 $\text{LPIPS}_r$ (Zhang et al., 2018) on the ImageNet validation set reconstruction. These results outperform Emu2's reconstruction performance and significantly surpass SEED-LLaMA and SEED-X (without conditional input). Fig. 5 visually illustrates our method's superior reconstruction quality.

### 4.2.2 SEMANTICS-GUIDED GENERATION

While fine-grained reconstruction is crucial for precise image manipulation, tasks like text-to-image generation benefit from a balance of semantic fidelity and output diversity. Our approach leverages coarse-grained features (such as $f_4$) to implement semantics-guided image generation that preserves diversity in outputs. To quantify this semantics-guided diversity, we decode twice to obtain two images from the same image input using different random seeds and measure their differences, denoted as $\text{PSNR}_d$ and $\text{LPIPS}_d$. Tab. 1 presents the diversity results for various visual decoding models and feature scales. Notably, our $f_3$ and $f_4$ scale decoders produce more diverse samples compared to the decoders in SEED-X and Emu2, while still preserving the core semantics of the input, as illustrated in Fig. 5. This demonstrates our approach's effectiveness in balancing semantic accuracy with generative diversity, a crucial factor in tasks like text-to-image generation.

$f_4$ scale-seed: 1    $f_4$ scale-seed: 2    $f_3$ scale-seed: 1    $f_3$ scale-seed: 2    Emu2-seed: 1    Emu2-seed: 2

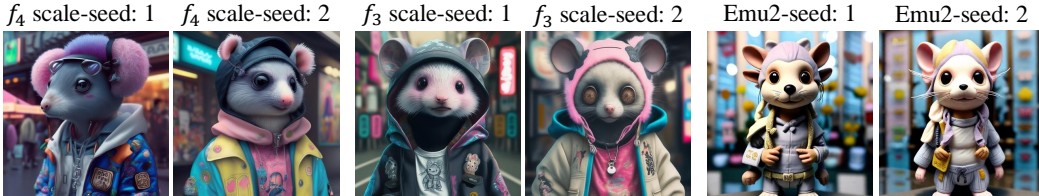

Generation prompt: Anthropomorphic rat, wearing harajuku street wear, decora kei, hyper realistic, clothing shops, shop signs, barbie aesthetic, kidcore, graphic.

Figure 6: Diversity visualization of text-to-image generation results from PUMA feature scales $f_4$ (1 visual token), $f_3$ (4 visual tokens), and Emu2 (Sun et al., 2024b). The generated features are input to corresponding diffusion-based decoders with different random seeds.

Table 2: Diverse text-to-image generation evaluation on MSCOCO 30K validation set. CLIP-I and CLIP-T measure the similarity between generated images and ground truth images or prompts. $\text{LPIPS}_d$ quantifies the difference between two images generated from the same prompt, reflecting generation diversity. 5-scale Max denotes selecting the image with the highest score among the 5 outputs and computes the average maximum value.

| Model | Token num. | CLIP-I↑ | CLIP-T↑ | $\text{LPIPS}_d$↑ |
|---|---|---|---|---|
| SD-v1.5 (2022) | - | 0.667 | 0.302 | 0.692 |
| DALL-E 2 (2022) | - | - | 0.314 | - |
| SDXL (2023) | - | 0.674 | 0.310 | 0.600 |
| DALL-E 3 (2023) | - | - | **0.320** | - |
| SEED-LLaMA (2023) | 32 | 0.682 | - | 0.652 |
| Emu (2023) | 64 | 0.656 | 0.286 | **0.700** |
| Emu2 (2024b) | 64 | 0.686 | 0.297 | 0.329 |
| SEED-X (2024b) | 64 | 0.729 | 0.314 | 0.493 |
| PUMA ($f_4$ scale) | 1 | 0.699 | 0.295 | 0.613 |
| PUMA ($f_3$ scale) | 4 | 0.703 | 0.300 | 0.558 |
| PUMA (5-scale Max) | - | **0.736** | 0.317 | - |

Table 3: Image editing evaluation on Emu-edit test benchmark (Sheynin et al., 2024). 5-scale Max denotes selecting the image with the highest score among the 5 outputs and computes the average maximum value.

| Model | CLIP-I↑ | CLIP-T↑ | DINO↑ |
|---|---|---|---|
| InstructPix2Pix (2023) | 0.834 | 0.219 | 0.762 |
| MagicBrush (2024a) | 0.838 | 0.222 | 0.776 |
| EMU-Edit (2024) | **0.859** | 0.231 | **0.819** |
| OmniGen (2024) | 0.836 | 0.233 | 0.804 |
| PUMA ($f_1$ scale) | 0.802 | 0.258 | 0.679 |
| PUMA ($f_0$ scale) | 0.840 | 0.264 | 0.784 |
| PUMA (5-scale Max) | 0.846 | **0.270** | 0.785 |

### 4.3 DIVERSE TEXT-TO-IMAGE GENERATION

Our method can generate diverse outputs by utilizing the coarse-grained feature ($f_4$ and $f_3$ scales). This capability enables our model to produce diverse images that correspond to text conditions. Fig. 6 demonstrates that when generating images with a fixed text prompt utilizing feature scales $f_4$ and $f_3$, our model achieves high generation diversity. It also shows that $f_4$ scale outputs exhibit higher diversity, while $f_3$ scale results demonstrate better consistency. In contrast, the generation results of Emu2 (Sun et al., 2024b) show low diversity. For qualitative evaluation, Fig. 7 presents visualizations of our model's text-to-image generation with various prompts. For quantitative results, we evaluate our model on the MSCOCO 30K validation dataset (Lin et al., 2014) and present the CLIP-I, CLIP-T, and $\text{LPIPS}_d$ in Tab. 2, which the former two metrics measures the consistency while $\text{LPIPS}_d$ measures generation diversity. Compared with recent works, our model demonstrates superior performance in generation quality, diversity, and prompt relevance.

### 4.4 IMAGE EDITING

To assess PUMA's image editing capabilities, we evaluated it on the Emu-Edit test benchmark (Sheynin et al., 2024). Tab. 3 presents the results using CLIP-I, CLIP-T, and DINO (Caron et al.,

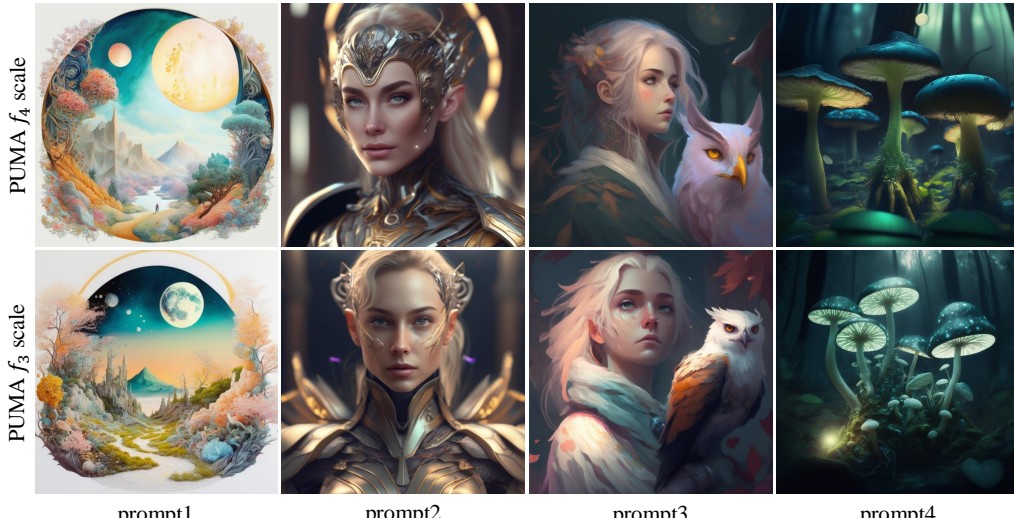

prompt1: Painting the entire universe in nutshell, line art drawing, magical scene, highly detailed, soft orange, mint green, soft blue, soft yellow, soft red, sharp outlines, sharp brush strokes, isolated.

prompt2: A beautiful blonde girl with futuristic wasp-inspired armour, compound eye, intricate design, unreal engine, cinematic lighting.

prompt3: A girl with white hair holding a harfang owl in her arms, artwork by james gilleard,vibrant colours.

prompt4: Cluster of magic mushrooms in a dark lush green forest during a storm.

Figure 7: Diversity visualization of text-to-image generation results from PUMA feature scales $f_4$ (1 visual token), $f_3$ (4 visual tokens), and Emu2 (Sun et al., 2024b). The generated features are input to corresponding diffusion-based decoders with different random seeds.

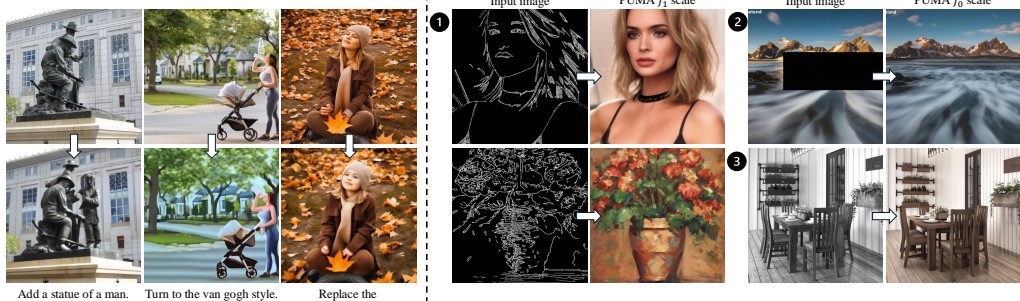

Figure 8: **Left:** Visualizations of PUMA's image editing result. Image editing utilizes $f_0$ scale feature to preserve the fine-grained detail of input image. **Right:** Visualization of PUMA's conditional generation results. ❶: canny-to-image generation; ❷: image inpainting; ❸: image colorization.

2021) scores. CLIP-I and DINO scores measure the model's ability to preserve elements from the source image, while CLIP-T reflects the consistency between the output image and the target caption. Our results demonstrate that PUMA exhibits strong preservation ability, second only to the current state-of-the-art model, EMU-Edit. Notably, PUMA achieves significantly better CLIP-T scores, even surpassing the state-of-the-art model. This indicates superior alignment between edited images and target captions. For qualitative evaluation, Fig. 8 provides visualizations of the editing results, illustrating PUMA's effectiveness in image manipulation tasks.

## 4.5 CONDITIONAL IMAGE GENERATION

We select a subset of canny-to-image, inpainting, and colorization tasks from the multigen-20M dataset to train PUMA's conditional image generation ability. Fig. 8 demonstrates the conditional generation results for these tasks. The $f_0$ feature scale results provide the highest preservation of

Figure 9: Comparison of $f_0$ and $f_1$ feature scales for tasks requiring precise controllability.

Table 4: Evaluation on multimodal understanding benchmarks. PUMA utilizes CLIP-Large encoder with $224 \times 224$ input. Und. and Gen. denote "understanding" and "generation", respectively.

| Type | Model | # Params | MMB↑ | MME↑ | GQA↑ | VQAv2$_{(test)}$↑ | POPE↑ | Vizwiz↑ |
|---|---|---|---|---|---|---|---|---|
| Und. Only | LLaVA-v1.5 (2024a) | 7B | 64.3 | 1510.7 | 62.0 | 78.5 | 85.9 | 50.0 |
| | InstructBLIP (2023) | 13B | - | 1212.8 | 49.5 | - | 78.9 | 33.4 |
| | Qwen-VL-Chat (2023) | 7B | - | 1487.5 | 57.5 | 78.2 | - | 38.9 |
| | mPLUG-Owl2 (2024b) | 7B | 64.5 | 1450.2 | 56.1 | 79.4 | 85.8 | 54.5 |
| Und. and Gen. | Emu (2023) | 13B | - | - | - | 57.2 | - | - |
| | NExT-GPT (2023) | 7B | 58.0 | - | - | 66.7 | - | 48.4 |
| | SEED-X (2024b) | 17B | 75.4 | 1457.0 | 47.9 | - | 84.2 | - |
| | Chameleon (2024) | 34B | - | - | - | 66.0 | - | - |
| | Emu2-Chat (2024b) | 40B | - | - | 65.1 | 84.9 | - | 54.9 |
| | PUMA (Ours) | 8B | 68.9 | 1490.3 | 60.6 | 76.2 | 85.2 | 47.9 |

image details, particularly for tasks like inpainting and colorization, while the $f_1$ scale offers better overall visual fidelity with limited generation diversity.

### 4.6 IMAGE UNDERSTANDING

We evaluate PUMA's image understanding performance on several MLLM benchmarks, including MMB (Liu et al., 2023), MME (Fu et al., 2024), GQA (Hudson & Manning, 2019), VQAv2 (Antol et al., 2015), POPE (Li et al., 2023), and Vizwiz (Gurari et al., 2018). Tab. 4 presents the results of this evaluation. Despite PUMA's relatively few 8B parameters and the use of an image encoder with $224 \times 224$ resolution input, it demonstrates competitive and often superior image understanding performance compared to other unified understanding and generation models. Notably, PUMA's performance on some metrics even surpasses that of understanding-only baselines. This performance can be attributed to PUMA's use of multi-granular continuous visual tokens as input to the MLLM. A detailed ablation study examining the impact of different scale features as input on image understanding tasks is provided in the Appendix, offering further insights into the effectiveness of PUMA's multi-granular approach.

### 4.7 ABLATION

We conduct an ablation study to examine the impact of feature scale selection on tasks requiring fine-grained controllability. Fig. 9 compares the outputs of $f_0$ and $f_1$ feature scales for image editing and colorization tasks. The results demonstrate that $f_1$ scale features are insufficient for preserving crucial image details, while $f_0$ scale features maintain the necessary fine-grained information for precise manipulation tasks. More ablation studies are in the Appendix.

### 5 CONCLUSION

In this paper, we introduce PUMA, a novel unified multi-granular MLLM that unifies various granular tasks in visual generation and understanding. By leveraging multi-granular representations, PUMA effectively addresses the challenge of balancing diversity and controllability in image generation tasks. Our approach demonstrates superior performance across a spectrum of visual tasks, including diverse text-to-image generation, image editing, inpainting, colorization, conditional generation, and understanding. PUMA's ability to adapt to varying granularity requirements within a single framework represents a significant advancement in MLLM capabilities. This work opens up new possibilities for more versatile and powerful multimodal AI systems, contributing to the broader goal of achieving artificial general intelligence in multimodal domains.

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

# A PUMA TRAINING

## A.1 VISUAL DECODING TRAINING

### A.1.1 DATASET DETAILS

For training the image decoding process, we leverage three large-scale datasets: Laion-2B (Schuhmann et al., 2022), Laion-Aesthetics (Burger, 2023), and JourneyDB (Sun et al., 2024a). To ensure high-quality generation capabilities, we apply a resolution-based filtering criterion, selecting only images with resolutions of $512 \times 512$ pixels or larger. We only use center crop as the data augmentation method.

### A.1.2 TRAINING SETTINGS

We train five dedicated image decoders for the $f_0$, $f_1$, $f_2$, $f_3$, and $f_4$ scale features respectively. The image encoder is the frozen CLIP-L image encoder (Radford et al., 2021). Each image decoder is initialized from the SDXL model. The VAE (Kingma, 2013) remains frozen throughout the training process. The corresponding image features are input to the diffusion model through the cross-attention mechanism, replacing the original text embedding input. We train the decoders using AdamW optimizer (Loshchilov, 2017) with a maximum learning rate of 8e-5, using linear learning rate decay and a gradient clipping value of 1.0. The training batch size is 1,024. The training steps for the five features are $40,000$, $30,000$, $20,000$, $15,000$, and $10,000$ respectively, with features containing more visual tokens using longer training steps. We use noise off value of 0.1 and random drop of 10% of the input image to blank image for classifier-free guidance.

## A.2 MLLM TRAINING

### A.2.1 TRAINING OBJECTIVE

PUMA employs a unified framework with supervision on both text tokens and image features. For text tokens, we use cross-entropy classification loss, while for image features, we adopt MSE regression loss. To balance the contribution of text and image outputs, we apply a loss ratio of 0.02 for text and 1.0 for image features. Within the image feature regression loss, we use different ratios for the progressively generated 5 scales of image features ($f_4$, $f_3$, $f_2$, $f_1$, and $f_0$), with ratios of 1024.0, 512.0, 64.0, 8.0, and 1.0 respectively. This scaling compensates for the varying number of tokens at each feature scale, with larger ratios for scales with fewer tokens. The training loss objective remains consistent across both the pretraining and instruction tuning phases.

### A.2.2 PRETRAINING DATASET DETAILS

During PUMA's pretraining phase, we utilize a diverse set of datasets including Laion-2B (Schuhmann et al., 2022), Laion-Aesthetics (Burger, 2023), GRIT (Peng et al., 2023), The Pile (Gao et al., 2020), OCR-VQA-200K (Mishra et al., 2019), and LLaVAR (Zhang et al., 2023b). For the image-text pair data in Laion-2B, Laion-Aesthetics, and GRIT, we randomly assign 50% of the samples to text-to-image training and 50% to image-to-text training, fostering both image generation and understanding capabilities. We employ center crop as the primary image augmentation technique. To train on the GRIT dataset for object grounding, we append 224 additional position tokens to the MLLM's codebook, representing object positions with bounding box coordinates [x_min, y_min, x_max, y_max]. We construct the training sequences by appending the tokens  and  to denote the beginning and end of each sequence. At the beginning and end of each image feature sequence, we include the special tokens [IMG] and [/IMG] to indicate the visual position.

### A.2.3 PRETRAINING SETTINGS

We conduct pretraining for 100K steps using the AdamW optimizer with a batch size of 2048. The maximum learning rates are set to 1e-4 for the projector and 3e-5 for the LLaMA backbone. We employ a 2,000-step warm-up period, cosine learning rate decay, and gradient clipping at 5.0 during pretraining. To optimize memory usage and computational efficiency, training is accelerated using DeepSpeed ZeRO Stage 3. The entire pretraining process is carried out on 256 NVIDIA V100 GPUs over a period of 10 days.

#### A.2.4 INSTRUCT TUNING SETTINGS

*High-quality Text-to-Image Generation:* We utilize Laion-Aesthetics (Burger, 2023) and JourneyDB (Sun et al., 2024a) with a data ratio 1:1 to instruct tune the text-to-image generation model based on the previous pretraining checkpoint. We use training batch size 2048 and train for 20,000 steps with the max learning rate 1e-5, warm up 1,000 steps, and cosine learning rate decay. Random crop with fixed aspect ratio is adopted as the image augmentation.

*Precise Image Manipulation:* We train the image manipulation task with SEED-Edit Ge et al. (2024a). It contains seven different operations: background alteration, comprehensive image changes, style alteration, object removal, object addition, localized modifications, and color/texture alterations. We train with batch size 1024 and train for 10,000 steps. The max learning rate is 1e-5, warm-up is 500 steps, and cosine learning rate decay is adopted. We apply random crop with fixed aspect ratio on the accordingly input image and output image. The sequence of the image manipulation sample is like "`[IMG]embedding of origin image[/IMG]instruct editing prompt[IMG]embedding of edited image[/IMG]`".

*Conditional Image Generation:* We train on the subset of MultiGen-20M dataset (Qin et al., 2023) including canny-to-image, image inpainting, and colorization. We use the training batch size $1,024$ and train for $20,000$ steps. The max learning rate is 1e-5, warm-up is 500 steps, and cosine learning rate decay is adopted. We apply center crop as the image augmentation. The sequence of the conditional image generation is like "`[IMG]embedding of origin image[/IMG]instruct conditional generation prompt[IMG]embedding of edited image[/IMG]`". The "`instruct conditional generation prompt`" contains the caption of the target image and with a 50% probability contain the task instruction like "`Please convert the canny image to a natural image`".

*Image Understanding:* We train image understanding task on the subset of LLaVA-OneVision (Li et al., 2024a) and Cambrain (Tong et al., 2024). Data about math/reasoning and cross-duplicated data in the two datasets are removed. We train with the batch size 512 and train all data for 1 epoch. The max learning rate is 1e-5 with the warm-up 500 steps. Cosine learning rate decay is adopted. We apply resizing as the image augmentation. Supervision is only applied to the output text tokens. We use the system message "`A chat between a curious user and an artificial intelligence assistant. The assistant gives helpful, detailed, and polite answers to the user's questions.`"

## B EVALUATION DETAILS

### B.1 IMAGE RECONSTRUCTION EVALUATION

To evaluate the reconstruction performance of different scales of features and our baselines, we use the ImageNet validation set, comprising 50,000 images. Each image is resized to a rectangular shape before being input into each image encoder. We assess reconstruction precision by computing $PSNR_r$ and $LPIPS_r$, which measure the difference between the reconstructed image and the original image.

Given the inherent randomness in the decoders, we measure reconstruction diversity by reconstructing each original image twice using different random seeds. We then calculate $PSNR_d$ and $LPIPS_d$ to quantify the difference between these two reconstructed images. Higher diversity is beneficial for downstream tasks such as text-to-image generation.

For PSNR and LPIPS evaluations, we use a resolution of $256 \times 256$ to align with the evaluation settings in previous works. For LPIPS evaluation specifically, we employ AlexNet as the feature extractor.

### B.2 TEXT-TO-IMAGE GENERATION EVALUATION

We evaluate text-to-image generation on the COCO 30K validation set (Lin et al., 2014). We use CLIP-I and CLIP-T scores to measure the consistency between the generated image and the ground truth image and caption, respectively. CLIP-Base-32 serves as the feature extractor for these metrics. To assess generation diversity, we calculate $LPIPS_d$ between two images generated using the same input prompt but different random seeds. The $LPIPS_d$ measurement details are consistent with those described in Sec. B.1.

Table 5: Ablation of different visual token input on image understanding. The experiments are conducted on LLaVA-v1.5 setting with CLIP-Large-224 visual encoder.

| Visual token type | Token number | MMB↑ | MME↑ | GQA↑ | VQAv2$_{\text{(test)}}$↑ |
|---|---|---|---|---|---|
| $f_4$ | 1 | 56.8 | 1252.6 | 0.0 | 64.1 |
| $f_3$ | 4 | 58.3 | 1285.5 | 0.0 | 67.0 |
| $f_2$ | 16 | 61.5 | 1403.0 | 46.6 | 71.1 |
| $f_1$ | 64 | 63.6 | 1400.8 | 58.4 | 74.4 |
| $f_0$ | 256 | **65.4** | **1464.9** | 58.8 | **76.9** |
| $f_4$-$f_0$ | 341 | 65.1 | 1445.5 | **61.0** | **76.9** |

### B.3 IMAGE EDITING EVALUATION

We evaluate image editing performance on the Emu-Edit benchmark Sheynin et al. (2024). To assess editing quality, we adopt CLIP-I, CLIP-T, and DINO scores. CLIP-I and DINO Caron et al. (2021) scores measure the model's ability to preserve elements from the source image, while CLIP-T reflects the consistency between the output image and the target caption. For the DINO score, we employ DINO-Small-16 as the feature extractor.

### B.4 IMAGE UNDERSTANDING EVALUATION

For image understanding tasks, we use the same evaluation setting as LLaVA-v1.5 (Liu et al., 2024a). During evaluation, we use the system message "A chat between a curious user and an artificial intelligence assistant. The assistant gives helpful, detailed, and polite answers to the user's questions."

## C ABLATION OF DIFFERENT SCALE FEATURES AS INPUT ON IMAGE UNDERSTANDING TASK

Given that PUMA adopts a unified multi-granular image feature as both input and output for the MLLM backbone, we conducted an ablation study to investigate the influence of different scales of image feature input on image understanding tasks. For a fair comparison, we adopted the standard LLaVA-1.5-7B pretraining and finetuning setting, only changing the image encoder to a 224-input CLIP-Large with different granularities of features.

Tab. 5 presents the results of this ablation study. The findings demonstrate that finer-grained features generally lead to better performance in image understanding tasks. Notably, utilizing all image features from $f_4$ to $f_0$ (the PUMA setting) achieves comparable performance to using all 256 visual tokens of the finest scale ($f_0$). These results validate that the unified visual input and output format of PUMA provides a robust foundation of visual features for image understanding tasks, effectively balancing performance across different granularities.

## D SELECTION OF 5 SCALE FEATURES IN TEXT-TO-IMAGE GENERATION

PUMA generates images at 5 granularity levels, allowing users to select the output that best meets their requirements. In our evaluation of diverse text-to-image generation, we produce 5 image outputs for each input prompt, corresponding to the 5 feature scales. To assess performance, we select the image with the highest CLIP-I and CLIP-T scores among the 5 outputs and compute the average maximum value. Tab. 6 presents the CLIP-I and CLIP-T scores for each of the 5 feature scales.

The results demonstrate that different granularity levels excel in various aspects of image generation. Notably, the ability to select the best output from multiple scales (PUMA 5-scale Max) yields significantly improved CLIP-I and CLIP-T scores compared to any single scale, highlighting the advantage of PUMA's multi-granular approach.

| PUMA ($f_4$) - 1 token | PUMA ($f_3$) - 4 tokens | PUMA ($f_2$) - 16 tokens | PUMA ($f_1$) - 64 tokens | PUMA ($f_0$) - 256 tokens |

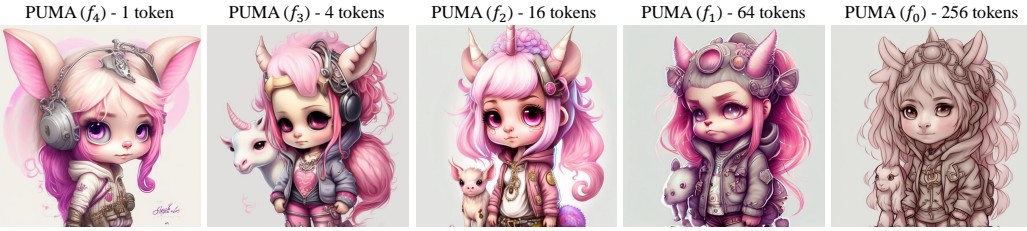

Generation prompt: Hyper realistic happy steampunk chibi girl wearing a pink hoodie with a pet on white background.

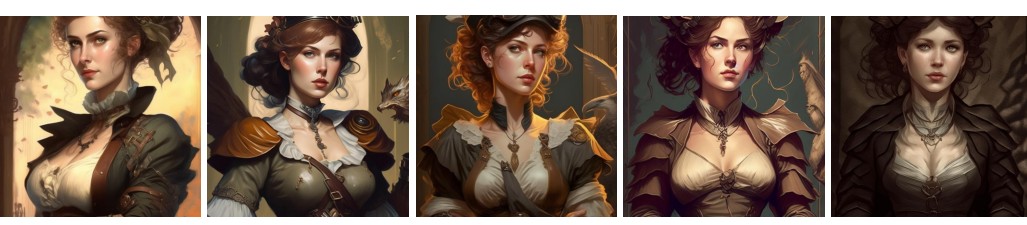

Generation prompt: Beautiful portrait by J.c. Leyendecker, beautiful lighting, Victorian Female Hunter, Fantasy.

Figure 10: Visualization of PUMA text-to-image outputs across five scale features given the generation prompt.

Table 6: CLIP-I and CLIP-T scores on MSCOCO 30K validation set with different feature scales.

| Model | Token num. | CLIP-I↑ | CLIP-T↑ |
|---|---|---|---|
| PUMA ($f_4$ scale) | 1 | 0.699 | 0.295 |
| PUMA ($f_3$ scale) | 4 | 0.703 | 0.300 |
| PUMA ($f_2$ scale) | 16 | 0.703 | 0.301 |
| PUMA ($f_1$ scale) | 64 | 0.693 | 0.299 |
| PUMA ($f_0$ scale) | 256 | 0.621 | 0.280 |
| PUMA (5-scale Max) | - | 0.736 | 0.317 |

## E    QUALITATIVE RESULTS OF TEXT-TO-IMAGE GENERATION ON FIVE SCALE FEATURES

In the text-to-image generation task, PUMA produces five distinct images corresponding to the five feature scales, all derived from a single input generation prompt. Fig. 10 presents samples of outputs across these five scales for given generation prompts.

## F    MORE QUALITATIVE RESULTS

We present more qualitative cases for image reconstruction, diverse text-to-image generation, editing, and conditional image generation, as shown in Figures 11 to 15.

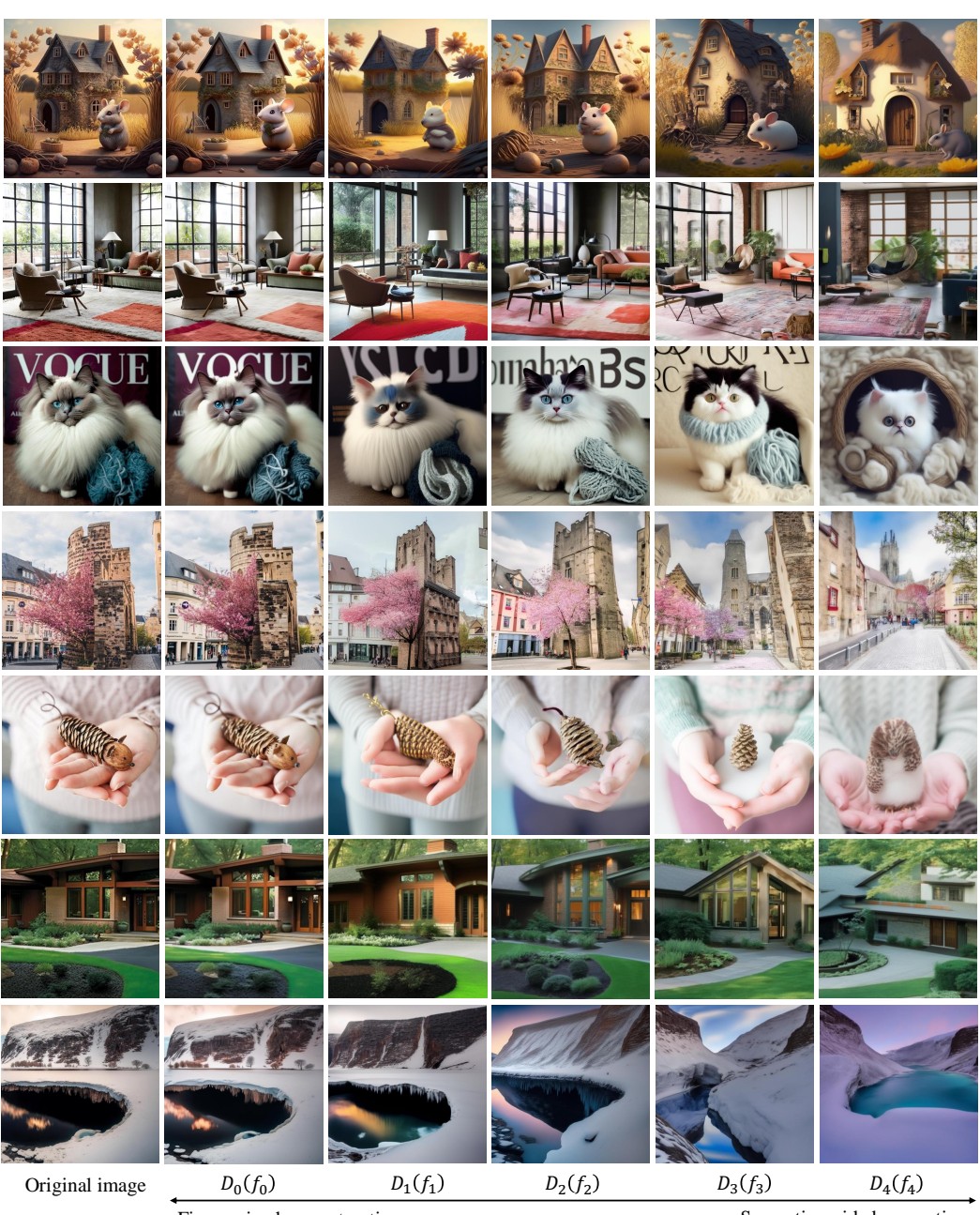

Figure 11: More visualizations on multi-granular visual decoding from fine-grained to coarse-grained granularity.

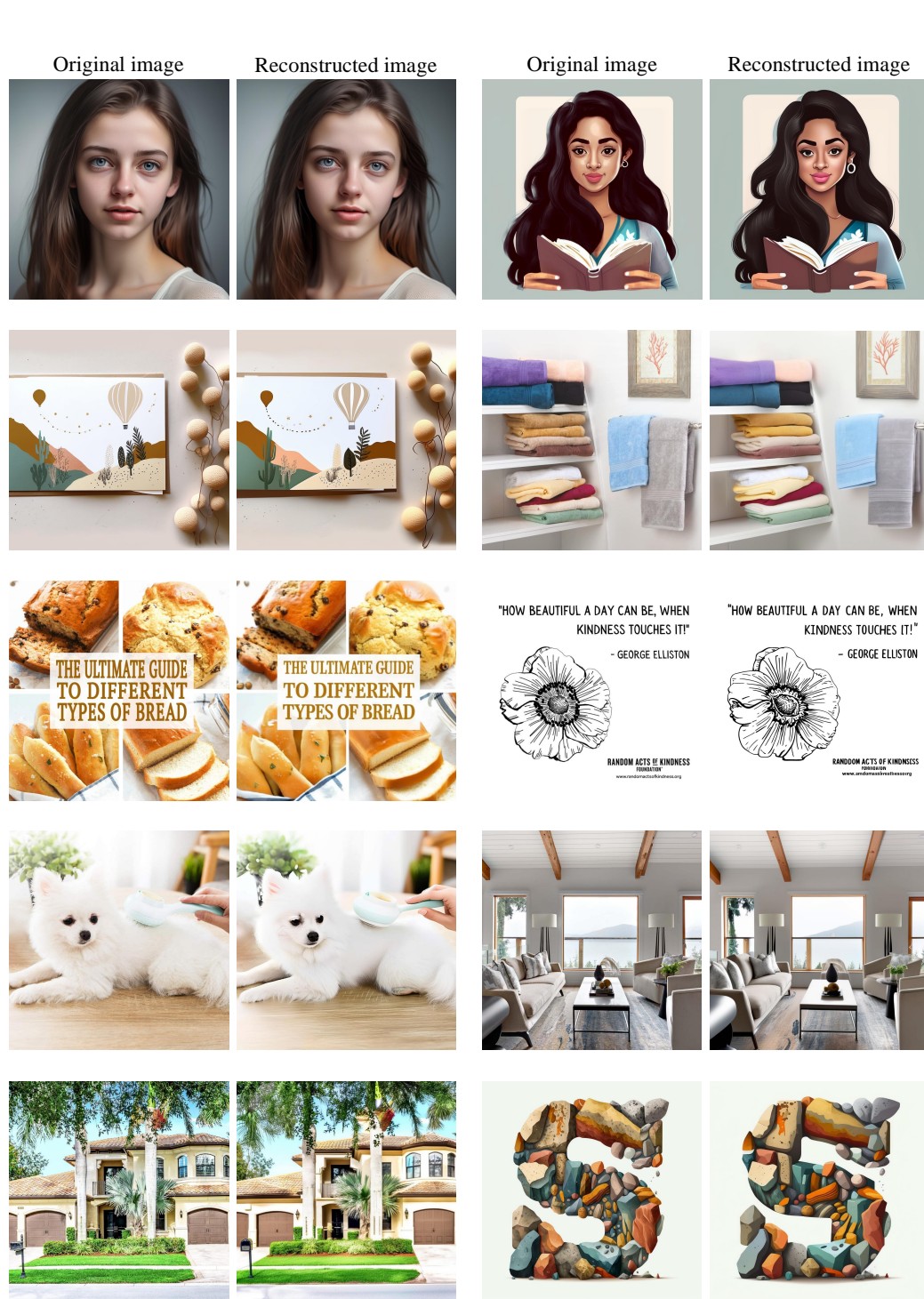

Figure 12: More visualizations on fine-grained image reconstruction with $f_0$ scale feature.

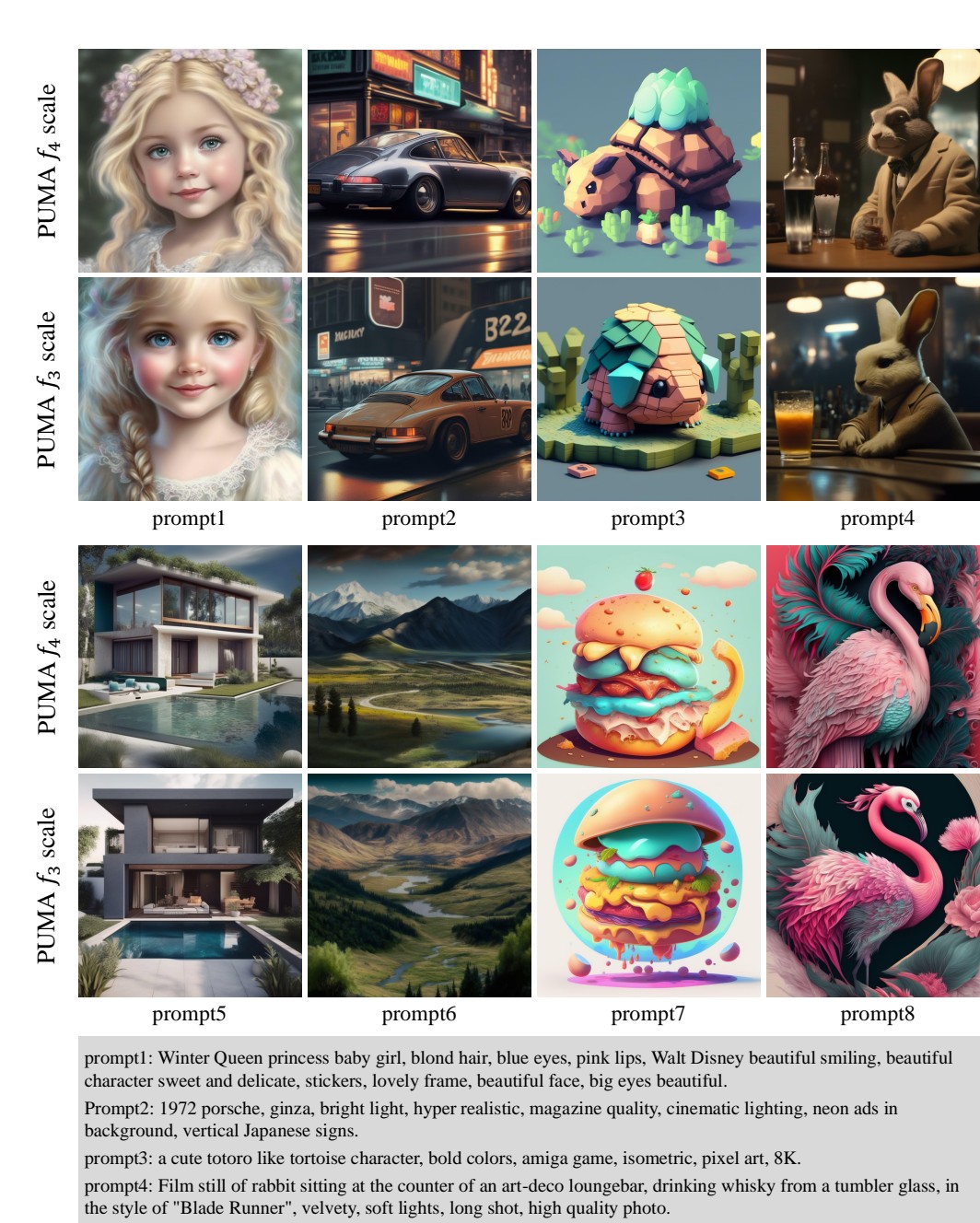

Figure 13: More visualizations on text-to-image generation utilizing $f_4$ and $f_3$ scales.

prompt1: Winter Queen princess baby girl, blond hair, blue eyes, pink lips, Walt Disney beautiful smiling, beautiful character sweet and delicate, stickers, lovely frame, beautiful face, big eyes beautiful.

Prompt2: 1972 porsche, ginza, bright light, hyper realistic, magazine quality, cinematic lighting, neon ads in background, vertical Japanese signs.

prompt3: a cute totoro like tortoise character, bold colors, amiga game, isometric, pixel art, 8K.

prompt4: Film still of rabbit sitting at the counter of an art-deco loungebar, drinking whisky from a tumbler glass, in the style of "Blade Runner", velvety, soft lights, long shot, high quality photo.

prompt5: a container designed compound built for a group home styled living space. 6000 SQ ft with 7 bedrooms and 1 adult suite. give the view landscape style with a smilling pool in the front.

prompt6: Open valley from mountains, aspen, hyper-realistic.

prompt7: Cartoon, pixar style, the planet hamburger, line art drawing, magical scene, highly detailed, soft orange, soft blue, soft pink, soft red, sharp outlines, sharp brush strokes.

prompt8: Beautiful colorful flower motif graphic, in the shape of an elegant flamingo in the style of Hayao Miyazaki, front view.

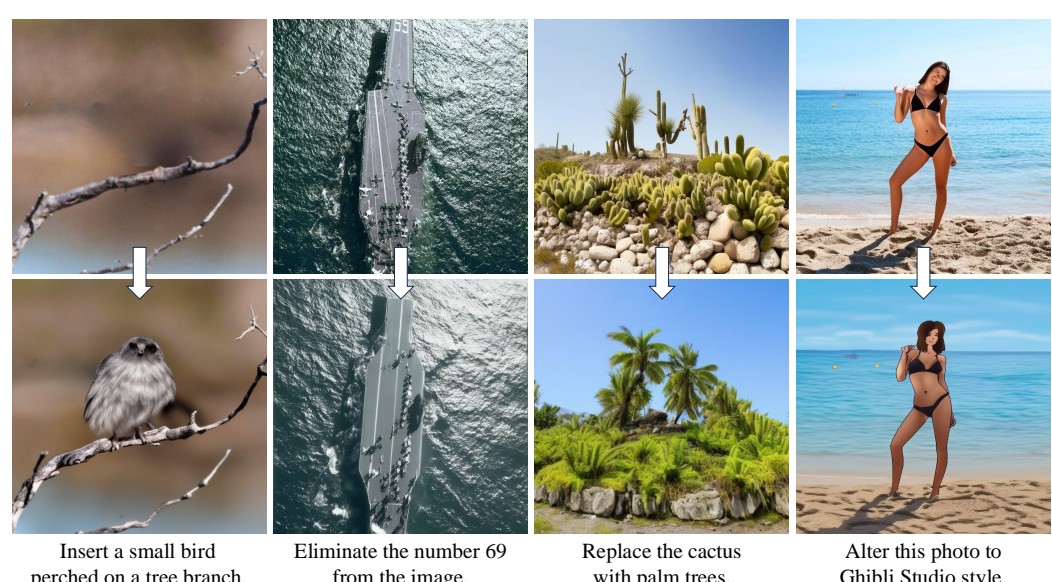

| Insert a small bird perched on a tree branch. | Eliminate the number 69 from the image. | Replace the cactus with palm trees. | Alter this photo to Ghibli Studio style. |

Figure 14: More visualizations on image editing.

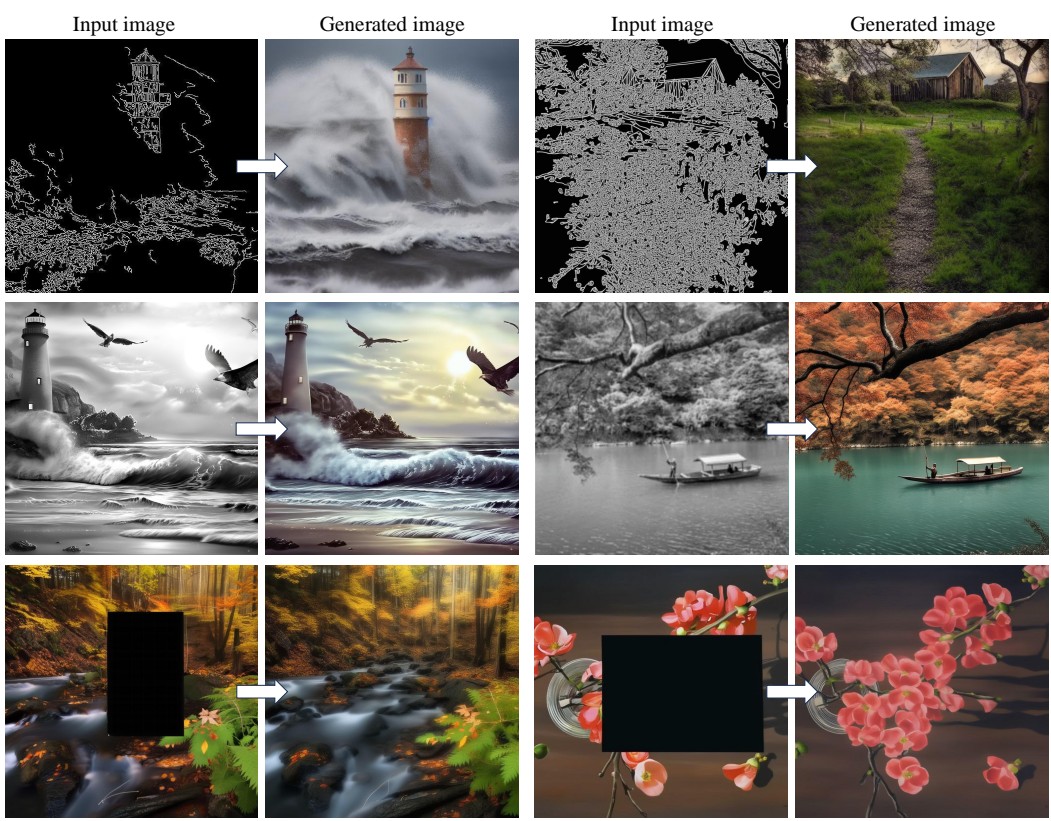

Figure 15: More visualizations on conditional image generation.

