# OpenReview forum: "PUMA: Empowering Unified MLLM with Multi-granular Visual Generation"
_ICLR.cc/2025/Conference — ICLR 2025 Conference Withdrawn Submission_

### Official Review · Reviewer_smH3 · 2024-11-01

**Soundness:** 2
**Presentation:** 3
**Contribution:** 2
**Rating:** 6
**Confidence:** 3

**Summary:**

This paper address the problem of achieving both strong control and strong diversity in image generation tasks powered by MLLM with diffusion model as image decoder.
To address this problem, the paper propose to represent image using multi granular embeddings, which conceptually is similar to an image pyramid, and train MLLM to predict embeddings at all scales in autoregressive manner. The predicted fine grained representation can be used in diffusion based image decoder to generate images following strong input condition, such as in image editing, while the gross grained one can be used to generate images with diversity.

**Strengths:**

The paper clearly stated the problem, which is a common challenge for MLLM that needs to balance the different needs of multiple visual generation tasks. Solving such challenge will definitely help improve the overall capability of the MLLMs.

The proposed solution is relatively simple (in a good way) and clean, and is in principle straightforward to implement. Experiments show that the proposed approach is also capable of solving the proposed problem, which helps both high diversity and high control needs.

The paper is clearly written with details on both training and evaluations in appendix. Shouldn't be too difficult to re-implement by 3rd party researchers.

**Weaknesses:**

The simple design will lead to a model that's slow in inference time:
a),  the image feature sequence length is essentially doubled with a lot of redundant information. Leading to a larger context window in inference time, and 2x inference cost as the output sequence length will be doubled for visual generation tasks.
b) several images has to be generated, each from a different diffusion decoder for the corresponding feature granularity. In the end, only one of these outputs will be used as the final output.

The result final model size will be much larger as well since there will be several SDXL UNet checkpoints, each is around 2B parameters.

In experiments, the paper fine-tunes separate MLLMs for different visual-linguistic tasks. This setup is inconsistent with the problem formulation, where one MLLM has to balance the needs of several tasks.

The authors did not discuss how to select the feature scale for final output. This is vaguely discussed in Appendix D. It should be more clearly discussed as it is an important part of the entire generation flow.

**Questions:**

The current feature pyramid is constructed based on spatially averaging features from previous finer levels.  This approach means information in f_4, f_5 is already represented in low level features f_0. I want to learn from the authors on their opinion towards a possibly more compact, top down representation where only information not present in f_5 are encoded in f_4 and recursively to lower levels.

In terms of generation diversity, there are existing approaches in diffusion models that can increase the output diversity. No need for experiments, but would be great to learn authors' opinion on using these approaches to address diversity needs, e.g. https://arxiv.org/abs/2310.17347, https://arxiv.org/abs/2310.12583

---

### Official Review · Reviewer_qD5J · 2024-11-01

**Soundness:** 2
**Presentation:** 3
**Contribution:** 2
**Rating:** 5
**Confidence:** 4

**Summary:**

This paper proposes a unified MLLM, PUMA, which uses multi-granularity visual features as input and output to handle the different granularity requirements of various image generation tasks.

**Strengths:**

1. The idea of multi-granularity for image generation is interesting.
2. The visualization of multi-granularity images is clear and the performance looks good.
3. The paper is well written and easy to follow.

**Weaknesses:**

1. The novelty of this paper seems limited; although it claims to introduce multi-granular visual generation, many papers (e.g., SEED-X[1], Matryoshka[2]) already focus on multi-granularity visual features, both in MLLM and image generation. Furthermore, the implementation of multi-granularity merely involves using different pooled visual features and different diffusion decoders, which also introduces additional parameters and computation.
2. In Table 1, is PUMA using (1+4+16+64+256) visual tokens? This number is much larger than in previous works. When using the same number of visual tokens, PUMA appears to underperform compared to other methods.
3. In Figure 2, the input and output of the PUMA Multimodal Generative Model are visual features covering different levels of visual granularity. I wonder whether the model can perform single granularity image generation during the inference stage (or human-specified multi-granularity).
4. In Tables 3 and Table 4, the authors should indicate how many visual tokens each model uses.


[1] Ge, Yuying, et al. "Seed-x: Multimodal models with unified multi-granularity comprehension and generation." arXiv preprint arXiv:2404.14396 (2024).
[2] Cai, Mu, et al. "Matryoshka Multimodal Models." arXiv preprint arXiv:2405.17430 (2024).

**Questions:**

Please refer to weaknesses.

---

### Official Review · Reviewer_QcPP · 2024-11-04

**Soundness:** 3
**Presentation:** 3
**Contribution:** 3
**Rating:** 5
**Confidence:** 3

**Summary:**

This paper proposes a method for multimodal visual generation and understanding at different levels of granularity. Concretely, the method proposes the use of image encodings at different resolutions combined with text tokens in a single autoregressive model called PUMA. Modeling image encodings at different granularities allows for more diverse image generation depending on the granularity of features from which the images are generated. In addition to the autoregressive model that generates the image encodings, there are multiple diffusion decoder models for each level of granularity in the generated encodings. In their experiments, the authors show the flexibility of using the different levels of generation granularity for different generation tasks such as text-to-image generation, editing, pinpointing, colorization and conditional generation. In addition, the authors show that the model is able to compete with other methods in understanding tasks

**Strengths:**

Originality:
To the best of my knowledge, this paper seems to be the first to combine multi-granularity image and text modeling. This makes it interesting and could spark some directions in the future.

Quality:
The thorough evaluations a presented in the experimental section are of fair quality which helps the reviewers make a better assessment of the contributions and significance of the paper.

Clarity:
The presentation of some of the ablations could use some work. Please see weaknesses section

Significance:
The significance of this paper given the results in somewhat average, but maybe clarifications and improved results could make it more significant?

**Weaknesses:**

* Understanding is negatively affected by multi-granularity modeling.
Looking at Table 5 in the appendix, it is not clear whether multi-granularity modeling adds to the understanding ability of the model. For half of the metrics the performance becomes worse, for one is the same, and only for a single metric the performance actually seems to improve. Do the authors have an intuition of why this is the case? Is spending part of the compute budget in the coarse level features the reason for this? Or is it the features themselves that confuse the model.


* Understanding ablations.
It is not clear how the ablations in appendix C were done. Did the authors train multiple models with encodings from only a single granularity? Or did the authors remove the encodings at other granularities at inference time to evaluate on how well the understanding performs when given a single granularity encodings?


* Reconstruction performance.
In the provided evaluations, the proposed method outperforms previous models in terms of reconstruction given image encodings at the finest granularity. Is this an effect of the multi-granularity learning? Or is it just an effect of the encoder / decoder combination. If it's just the encoder / decoder combination, does this result validate the proposed multi-granular idea of the paper?



* Use of multiple decoders.
Is using multiple decoders necessary? Did the authors try using a conditioning signal to tell the diffusion decoders what type of inputs are being fed for generation?

**Questions:**

Please see weaknesses

---

### Official Review · Reviewer_1LzQ · 2024-11-04

**Soundness:** 4
**Presentation:** 4
**Contribution:** 1
**Rating:** 3
**Confidence:** 5

**Summary:**

The paper proposes a model that combines generation and understanding tasks, referred to as PUMA. The proposed model extracts features at multiple scales from a CLIP model and trains an array of diffusion models conditioned on scale-specific features. A multimodal LLM is optimized then to generate either text tokens or image features jointly using standard next word prediction or L2 distances between features (Eq 2).  The proposed model is pre-trained on an array of tasks that involve text-to-image and image-to-text generation. Then four dedicated models are fine-tuned and evaluated on four downstream tasks (Sec 3.4).

**Strengths:**

* The idea of multiscale features in the image generation domain for targeting a diverse array of tasks is original and useful. Some features at some scale might benefit one task while other features might be beneficial for other tasks as shown in some of the experiments.
* The paper makes a compelling argument for the need of unifying generation and understanding and represents an attempt in this direction.
* The paper is well written and well organized in general.

**Weaknesses:**

* The main concerns is that the paper does not fulfill what seems to be its main promise: To have a single integrated model that can perform generative and understanding tasks. The models that are evaluated for each of the four tasks are different model checkpoints that are produced by targeting each downstream task specifically through different finetuning datasets.
* I also have some concern with not including FID (Frechet Inception Distance) as is customary to evaluate generative tasks. While this metric is not perfect it seems the paper is currently bypassing this measure of quality and mostly focusing on diversity and adherence to prompt.
* There are a couple of comparisons that could have been included to other recent models that provide evaluation results for the same checkpoint across understanding and generation and/or provide publicly available checkpoints for evaluation such as GILL (Koh et al NeurIPS 2023) -- mentioned in the paper, Unified-IO 2 (Lu et al CVPR 2024), and GenLLava (Hernandez et al 2024). These models are not finetuned for each type of tasks, generation, editing or understanding but are finetuned jointly to perform well across types of tasks.

**Questions:**

* I don't have any questions. I think addressing my first point would require major new results which would be only appropriate for a new submission. Unless I missed something major I am not sure how would this be addressed in the rebuttal. Perhaps a basic follow up to improve the paper would also be evaluating the pre-training performance of the presented model.
* I think however the multi-scale image feature contribution to be the most interesting part but does not seem to quite fit in the premise of the paper of having a unified generation and understanding model but seems a rather orthogonal contribution. If this was to be the main contribution of the paper, then the paper would need to be re-framed in terms of this contribution and then more ablations would be needed or a way to automatically select the feature scale as right now the best results are obtained through PUMA ( max of 5 scales) oracle performance.

---

### Note · Authors · 2024-11-14

**Comment:**

Thanks so much for all the reviewers' suggestions. We have decided to withdraw the submission.

**Withdrawal Confirmation:**

I have read and agree with the venue's withdrawal policy on behalf of myself and my co-authors.